# By Internal Network or by External Network?—Study on the Social Network Mechanism of Reducing the Perception of Old-Age Support Risks of Rural Elders in China

**DOI:** 10.3390/ijerph192215289

**Published:** 2022-11-19

**Authors:** Jianliang Nie, Rong Fan, Yufeng Wu, Dan Li

**Affiliations:** School of Public Administration, Northwest University, Xi’an 710127, China

**Keywords:** social networks, internal network, external network, rural elders, perception of old-age support risks

## Abstract

Nowadays, it is a general trend for China to enter a deep aging society, and the aging situation of the rural population is particularly severe. As informal endogenous resources in rural areas, social networks play an essential role in ensuring elders’ later life. Data were drawn from a questionnaire survey of 1126 rural elders in 11 provinces of China. Descriptive statistics and an ordinary least square regression model were conducted to explore the impact of social networks on the perception of old-age support risks of rural elders. There was a significant positive association between the social network size and the reduction in perception of old-age support risks of rural elders. The reduction effect was mainly reflected in the internal network size, whereas it was not evident in the external network size. There was a significant positive association between the heterogeneity of the network and the perceived level of old-age support risks of rural elders. There was a significant positive association between the communication frequency of external network relationships and the perceived level of old-age support risks of rural elders. We found a significant negative association between the ratio of communication frequency between the internal and external network relationships and the perception of old-age support risks of rural elders. Compared with the external network, the internal network had a more evident reduction effect on the social network mechanism of perception of old-age support risks of rural elders.

## 1. Introduction

According to the Seventh National Population Census released by the National Bureau of Statistics of China, the proportion of people aged 65 and above in China accounted for 13.50% of the total population in 2020, and the ratio increased by 4.63%, compared with the Sixth National Population Census in 2010. In addition, the “post-60s” group born in the second baby boom entered retirement age, and the growth rate of elderly population is further accelerating, indicating a general trend for China to enter a deep ageing society. At the same time, a large outflow of adult labor from the countryside has further aggravated the rural aging problem, so that the aging of China presents the apparent characteristics of urban–rural inversion. According to the “2020 National Bulletin on the Development of Undertakings for the Aged”, the proportion of elders aged 65 and above in rural areas is 17.72%, 6.61% higher than elders in urban areas. As such, the “elders’ support” of the rural elders has gradually become a hot issue that has aroused extensive attention [1].

The interaction between social networks and the later life of elders is complex and dynamic. Social networks play a significant role in shaping the nature and the quality of elders’ later life [2]. In the traditional Chinese rural society, there is a complex network of acquaintances and simple mutual benefit, trust, and cooperation among villagers. Older adult individuals in the village have a self-centered social network, and individuals in the network provide a series of necessary support for the well-being of their internal members by consciously or unconsciously assuming the responsibility for elders [3]. Although the rural elders can exploit the mobilization of resources in social networks to obtain certain social support to reduce old-age support risks, their interpersonal and social networks have shown significant changes in many aspects, because of the rural labor outflow of the population, their occupational exit, etc. [4,5]. In addition, the emergence of the core and empty-nest trend in the traditional rustic family structure makes the rural social networks looser. In this context, social networks cannot be ignored, contributing to the reduction in the old-age support risks of rural elders and their perception of old-age support risks. However, the association is not well understood among the rural elders in China. Here, comes the question: do the rural elders depend on the internal network or the external network to reduce their perception of old-age support risks?

Plenty of problems may be more common among the rural elders who face physical disability, psychological stress, and economic cost barriers caused by spatial transformation, accompanied by the characteristics of increased health risks as they age, low education levels, low income, and lack of companionship. For example, Chen believed that the emergence of old-age support risks was closely related to changes in rural social structure, the decline of the traditional old-age support model, and the infiltration of urbanization risks into rural areas [6]. In the era of low fertility, old-age support risks would be exacerbated by rural–urban migration, continuous low fertility rates, and the extension of life expectancy [7]. Family support is of the utmost importance for preventing old-age support risks for rural elders [8]. Especially in rural China, due to the general lack of formal social support system and small coverage, the rural elderly mainly rely on their families for financial or nursing support [9]. With the changes in family structure in recent years, the “daughter bears the primary responsibility for supporting her parents” has gradually developed into a new component of China’s rural old-age support order [10,11], which can not only reduce the worries of the rural elders, but also enhance the adaptability of the family old-age support model. However, the changes in traditional agricultural society brought a shock to the foundation of rural family old-age support, making social old-age support become the core way of future old-age support model in rural China [12]. However, there is a relative lack of research regarding the social network mechanism of rural elders in China. More attention is required, in particular, the relationship between social network mechanism and their perception of old-age support risks.

Diversified old-age support strategies, especially social networks, play essential roles in reducing old-age support risks for rural elders. As an important carrier of social capital, the social network is a relatively stable relationship system formed by the interaction between individual actors in society [13]. According to the Social Convoy Model, individuals would experience different life-courses in pace with changes in their subjective and objective circumstances, and they would constantly obtain support from social networks in different stages to escort their life based on life-course and functional perspectives [14]. Due to the decline of the physical function of rural elders and the deterioration of their children’s old-age support function caused by the migration of their adult children, social networks have become an important survival strategy for the rural elders who are in the end stage of the lifespan, to rely on in the future old-age support [15]. The rich social capital in social networks can bring social support network benefits such as information, material resources, spiritual comfort, and social support, benefiting the old-age security function for the rural elders by improving material conditions, lack of companionship, respect for needs, and other forms [16].

A social network can be referred to as a composite characterization of the main components of an individual’s social network relationships and a comprehensive description of their interpersonal environment [17]. The construct of a social network can be generally defined as social capital, which reflects the collection of interpersonal contacts that people variously maintain and which provide them with access to social, emotional and practical support [18]. Studies have reported on the structural attributes (e.g., network member’s composition, size and contact frequency) or measurement scales (e.g., ‘family’, ‘friends’, ‘diverse’ and ‘restricted’ networks) to identify different social networks [19,20]. Social networks could be divided into family networks and friend networks based on kinship [21]. Among them, family networks has the “reverse structure” in age sequences and has very important effects on the well-being of the rural elderly [22]. Family networks could also improve life satisfaction for the elderly by decreasing the level of physical abuse they experience [23]. Older adults’ interactions with their friend networks could enhance their sense of self-worth, improve negative psychological states [24,25], and enhance the health status [19]. In addition, much research has proved that intimate interactions between older adults and members of a social network contributed to relieving stress, reducing depression levels, and promoting physical and mental health [26,27,28], and in turn improved their quality of life [29,30]. Furthermore, several studies have explored the effects of different types of social networks on cognitive function [31], well-being [32] and depressive symptoms [33] in older adults. In terms of the social network mechanism, evidence has demonstrated the inter-relationships of both the causal mechanisms by which social network influences physical health and the ways in which this influence can inform potential intervention strategies [34]. In the field of old-age support risk, most studies mainly focused on the overall old-age support risks faced by the elderly [6,7,35]. Only a few studies have explored its impact on the perception of old-age support risk from the perspectives of family security [36], social security [37] and offspring support [38]. However, evidence on the association between the social network mechanism and the perception of old-age support risk among rural elders still lacked support from a wider range of data. As an external manifestation of old-age support risks, the perception of old-age support risks should be taken seriously [39]. Further exploration is needed to better understand how an external social network is related to old-age support risk.

Therefore, this paper used the questionnaire data of 1126 rural elders in 11 provinces of China to explore the reduction effect of social networks on the perception of old-age support risks of rural elders from the perspective of the size, heterogeneity and communication frequency of the social network. To expand the research depth, this paper also investigated the impact of social networks and other variables on the perception of old-age support risks of different rural elder groups.

## 2. Materials and Methods

### 2.1. Data

The data were collected from a sample survey of older adults aged 60 and above in 11 representative provinces in Eastern, Central and Western China. The survey was conducted by the Social Security Research Group of Northwest University in China, which is funded by the project of the National Social Science Foundation. The questionnaire was designed based on large-scale scientific surveys and relevant studies at home and abroad; then, a trial survey was conducted. We modified and optimized the questionnaires according to comprehensive consideration of results of the trial survey, the validity and the reliability. Secondly, the research group randomly selected 32 highly representative villages from 11 provinces according to the socioeconomic development level of different regions, including four provinces in the eastern region (Hebei, Fujian, Jiangsu and Shandong), four provinces in the central region (Shanxi, Henan, Hubei and Hunan), and three provinces in the western region (Shaanxi, Yunnan and Xinjiang). Finally, the survey randomly selected about 35 older adults aged 60 and above from each village, and a questionnaire survey was conducted through a face-to-face interview. One of the sample villages was a non-traditional village, which could not meet the sampling requirements well, and the actual sample size was small, so 1126 valid questionnaires from 31 villages were retained.

### 2.2. Measurement

The dependent variable of this study was the perception of old-age support risks of elders in rural areas. Based on the previous studies [40], the perception of old-age support risks can be measured by three dimensions of risk perception of physical disability, risk perception of economic dependence, and risk perception of absence of companionship. 

The following three dependent variables were included and presented:

Question 1: Are you worried about not being able to care for yourself in the future?

Question 2: Are you worried about becoming utterly dependent on others for financial support?

Question 3: Are you worried about being lonely without any company?

The options were, respectively, assigned as “very worried” = 5, “relatively worried” = 4, “average” = 3, “not too worried” = 2, and “not worried at all” = 1. A higher score indicates a higher perceived level of old-age support risks of the rural elders, and vice versa.

In order to measure the perceived degree of old-age support risks of the rural elders on the whole, a principal component analysis (PCA) was used to carry out factor analysis on these three dimensions and extract a common factor. Factor analysis is a kind of multivariate statistical analysis method in which some variables with complex relationships are attributed to a few unrelated comprehensive factors based on the internal dependence of the index correlation matrix [41]. PCA is a statistical method for data dimensionality reduction [42]. It converts high-dimensional variables into low-dimensional orthogonal variables, and the converted variables are called principal components [43]. PCA can provide initial solutions for factor analysis. In the factor analysis, Kaiser-Meyer-Olkin (KMO) was 0.725, the Bartlett sphericity test reached the significance level of 0.00, and the cumulative variance percentage of the extracted common factors was 76.589%. The factor analysis result was satisfactory. To achieve better analysis results, the common factors were converted into exponentials. Index = (factor value + B) × A; A = 99/(maximum factor − minimum factor); B = (maximum factor − minimum factor)/99 − minimum factor.

### 2.3. Predictors

We drew references from social network size developed by Lubben et al. to measure the network size of the rural elders in our study [21]. A questionnaire survey was conducted to investigate and analyze how many family members or friends rural elders could meet, talk about personal matters with, and be offered help by. Six categories of numbers were assigned, with the value of 0 if the respondent answered “no”, the value of 1 if the respondent answered “1”, the value of 2 if the respondent answered “2”, the value of 3 if the respondent answered “3~4”, the value of 5 if the respondent answered “5~8”, and the value of 9 if the respondent answered “9 or higher”. In the measurement, the former three problems in family communication were indicators of internal network size; after three questions of friends as indicators of external network size, the sum two was used as the indicators of network size. Then, adding up the scores for the six categories, the score of internal network size and external network size both ranged from 0 to 27, the total score of the network size ranged from 0 to 54, and a higher score indicated their larger network size.

The network heterogeneity was measured by 12 questions about difficulties that the interviewees may encounter in daily life, and who they would ask for help when they encountered these problems (see Table 1). They could turn to village cadres, acquaintances, neighbors, friends, brothers, sisters, direct relatives of spouses, other close relatives, family members, other distant relatives, other relations and another 11 kinds for help. Based on the query results, whenever an object appeared, it was assigned a value of 1, while objects that never appeared were assigned a value of 0, then added up to obtain the variable of the network heterogeneity. The scope ranged from 0 to 11. The higher the score, the higher the heterogeneity of the network.

Our study defined three variables of the communication frequency of internal network relationships, the communication frequency of external network relationships, and the ratio of communication frequency between internal and external network relationships by “interaction frequency”. Among them, “interaction” mainly included family members, relatives, neighbors, friends and others in the same village. The internal network relationship was defined as the close relationship with their family, relatives and members of their family, whereas the external network relation was defined as the relationship with neighbors, friends and others in the same village. The value was assigned as “never” = 1, “rarely” = 2, “sometimes” = 3, and “often” = 4. We added them up to obtain the variables of the communication frequency of internal network relationships, and the communication frequency of external network relationships. The scope ranged from 3 to 12. To more precisely measure the relative strength of the communication between the rural elders and their network, the ratio of the communication frequency between the internal and external network relationships was created. The mean value of 1.091, which was greater than 1, indicated that the intensity of the communication relationship between the rural elders and the internal network members was higher than that of the external network members.

Demographic variables included gender, age, self-rated health status, mental health status, changes in health status, years of education and using mobile phones, pension benefits evaluation, annual household income level, whether the village has old-age support service places, number of children and living style, the terrain of the village and the region in this study. In particular, the variable of mental health status was the reverse measurement of depression or depression in the questionnaire, and the original values were “never” = 1, “rarely” = 2, “sometimes” = 3, “often” = 4, “always” = 5 were reassigned as “very bad” = 1, “poor” = 2, “general” = 3, “good” = 4, and “very good” = 5. The descriptive statistical results of the variables are shown in Table 2.

### 2.4. Statistical Analysis

Since the dependent variable, the perception of old-age support risks of elders, was a continuous variable, our study used the ordinary least square (OLS) regression model to analyze the factors affecting the social networks of rural elders. Regression coefficients and standard errors were reported in the regression results. A five percent significance level was chosen as a threshold for the inclusion of the model variables. All statistical analyses were performed using SPSS 26.0. The regression model was set as follows:(1)$$y=b_{0}+b_{1}x_{1}+b_{2}x_{2}+\ldots b_{n}x_{n}+\varepsilon$$

In Equation (1), y presents the perception of old-age support risks of elders; $x_{1}$ presents the independent variable; $x_{2}$ means the control variable; *b*_1_, *b*_2_, … *b_n_* present regression coefficients, respectively; and ε represents error items. Furthermore, on the basis of the OLS regression model, we discussed the influence of the social network and other variables on the perception of old-age support risks in different groups through the clustering estimation. Similarly, our study adopted the method of replacement of the dependent variable for the robustness test in the end, which is convenient for enhancing the explanatory power of the model.

## 3. Results

As Table 3 shows, Model 1 mainly showed the impact of network size, network heterogeneity, the communication frequency of internal network relationships, and the communication frequency of external network relationships on rural elders’ perception of old-age support risks. After controlling other variables, there was a significant statistically negative association between the network size and the perception of old-age support risks of rural elders. That is, a larger the network size indicated a lower perceived level of rural elders. Network heterogeneity had a significant positive impact on the perception of old-age support risks. There was no significant association between the internal network relationship frequency and their perception of old-age support risks. There was a significant positive association between the external network relationship frequency and their perception of old-age support risks.

Model 2 showed the impact of internal network size, external network size, network heterogeneity, the communication frequency of internal network relationships, and the communication frequency of external network relationships on the rural elders’ perception of old-age support risks. The first variable was the size of the internal network. Model 2 showed that the size of the internal network significantly negatively affected the perceived level of old-age support risks of the rural elders. In comparison, there was no significant association between the external network size and the perception of old-age support risks. The impact of three independent variables, namely network heterogeneity and communication frequency of internal/external relationships, on the perceived level of old-age support risks of rural elders was consistent with Model 1, which verified the previous research results again.

Model 3 and Model 4 mainly showed the impact of the ratio of communication frequency between internal and external network relationships on the perception of old-age support risks. All the models showed that the ratio of communication frequency between internal and external network relationships was negatively correlated with the perception of old-age support risks. It indicated that rural elders interacted more frequently with close internal members of the social network than external members, and their essential endowment resources were more abundant. Their perception of old-age support risks would also be reduced. The results of other independent variables in Model 3 and Model 4 were consistent with those in Model 1 and Model 2.

As Table 3 shows, there were significant associations between gender (male), age, self-rated health status (good and very good), mental health status (very good), pension benefit evaluation, number of children, whether the village has old-age support service places (have), the terrain of the village (plain), and the region (east) on the perception of old-age support risks. Among them, only the number of children presented a positive effect, and other variables had a negative effect on their perception of old-age support risks. Rural elders with more children had more limited energy to take care of their children, leading to a weaker parent–child relationship; that is, children were reluctant to take responsibility for supporting their parents when they became adults.

As Table 4 shows, the clustering estimation results of social networks’ perception of old-age support risks for rural elders were reported. Gender (Model 5 and Model 6), age (Model 7 and Model 8), and education (Model 9 and Model 10) were highly related to the individual characteristics of the rural elders, and they were deliberately selected as the group estimation standards. According to international standards and research data, the rural elders between 60 and 69 as the young elders, and farmers aged 70 and above in the village were classified as middle-aged elders [44]. According to the number of years of education, the rural elders were divided into uneducated and educated groups. The size of the internal network could significantly reduce the perception of old-age support risks for males (Model 5) and females (Model 6), young elders (Model 7), middle-aged elders (Model 8), and rural elders with various levels of education (Model 9 and Model 10), which indicated that family security was always the primary choice for rural elders. The influence of external network size was not significant among different populations.

However, the network heterogeneity and the ratio of communication frequency between internal and external network relationships had different effects on the perception of old-age support risks of rural elders with different genders (Model 5 and Model 6), ages (Model 7 and Model 8) and educational attainment (Model 9 and Model 10). There was a positive association between network heterogeneity and the variables of males (Model 5), females (Model 6), middle-aged elders (Model 8) and educated rural elders (Model 10). Meanwhile, the ratio of communication frequency between internal and external network relationships had a significant negative influence on females (Model 6), middle-aged elders (Model 8) and uneducated rural elders (Model 9).

Table 5 presents the results of the robustness test to facilitate the analysis and enhance the explanatory power of the model. We adopted the method of replacement of the dependent variable for the robustness test. The dependent variable of perception of old-age support risks extracted from the principal component method of factor analysis was replaced by the sum of the risk of physical disability, the risk of economic dependence, and the risk of absence of companionship. Table 5 shows that the network size, internal network size, and the ratio of communication frequency between internal and external network relationships had significant negative impacts on the perceived level of old-age support risks. There was no significant association between the external network size and risk perception. There was no significant association between the communication frequency of internal network relationships and risk perception, while the heterogeneity of the network and the communication frequency of external network interaction still had significant positive impacts on the perception of old-age support risks. This result was very consistent with the OLS regression model results in Table 3, indicating that this result was relatively ideal.

## 4. Discussion

This novel study explored the influence of social networks on the perception of old-age support risks of rural elders in China, the country with the largest scale of rural elders, based on a questionnaire survey of 1126 rural elders in 11 provinces in China. The innovation of this study lies in that, unlike previous studies that only explored the influence of social network size [45,46], social network type [31,32,33], family networks, and friend networks [47,48] on the elderly, our study not only explored the size and heterogeneity of social networks, but also the impact of internal and external networks on reducing the perception of old-age support risks among rural elderly was quantitatively measured in terms of the size and the communication frequency. This study provided an empirical reference for related research on the relationship between social networks and the reduced level of perception of old-age support risks.

Our findings indicated that a larger network size was associated with a lower perception of old-age support risks among rural elders. Consistent with previous studies [49,50], with a larger size of social network, more actors make contact with individuals, there are more channels for rural elders to obtain information, and there are more diverse social resources they can access, which can contribute to coping with support risks. From the perspective of network size, the social network can be divided into the internal network and the external network. The internal network size could significantly reduce the level of perception of old-age support risks, while the external network size had no significant impact. This was not to say that the internal network was superior to the external network, but instead the importance of the internal network in the interpretation of findings across studies. As mentioned above, internal network size mainly referred to the family network size dominated by family members. The family network was a community unit connected by blood relationships. It was embedded in the internal network to provide the most basic old-age support resources for the rural elders based on the most natural advantages, so as to reduce the level of perception of old-age support risks. Chinese families emphasized the interdependence of family members [51]. In this regard, our study echoed previous studies in highlighting the need for friends and neighbors; specifically, in traditional Chinese culture, friends and neighbors played a far less important role in providing support in old age [52].

With the accelerating urbanization in China, a large number of the rural adult labor force are leaving their hometowns and choosing to work and settle in cities [53,54]. Although the substantive size of the family network had been reduced by the mobility of adult children, its role was still influential, and the friend network was not a substitute [55]. Therefore, we need to maintain and expand the scale of the rural elders’ family network. On the one hand, we identified an urgent need to provide a corresponding policy and financial support, including building a family policy system focusing on support and protection, and encouraging migrant adult children to return to their hometown for employment by providing vocational skills training or subsidies. On the other hand, with China’s leap into the information age, more intelligent ways should be explored to reduce the perception of old-age support risks of elders in rural areas. For example, we should help elderly people in rural areas learn to use smart phones and other internet devices to enhance communication with their children and ease their perception of the risk of companionship loss.

We found that the communication frequency of internal network relationships had no significant impact on the perception of old-age support risks of the rural elders, while the communication frequency of external network relationships had a significant positive impact. The internal network was oriented to the situation that the rural elders have close relationships with their immediate relatives, relatives, and other family members. As an extension of the nuclear family network, the internal network played an essential role in ensuring the security of the elderly. In line with other studies [39], when adult children in rural areas went out for work, the closer elders were connected with internal network members, and the more secure their old age life would be. Therefore, the communication frequency of internal network relationships could not improve the perceived level of old-age support risks of the rural elders. We noted that the continued evidence of reduced old-age support risks required greater consideration of the social network mechanisms by which this reduction persists. Some study provided a good explanation of the reduction in the perception of old-age support risks, especially when the frequency of interaction between the rural elders and the members of the internal network was higher than the frequency of interaction with the members of the external network; they could more easily obtain the required old-age support resources through the stronger interpersonal influence and broader reciprocity of the members of the internal network [56].

In the results of clustering estimation, the ratio of communication frequency between internal and external network relationships had a significant negative influence mainly on female, middle-aged and uneducated rural elders. In rural society, men usually go out for work, while women mostly stay in rural areas to care for their families and children. The limitation of spatial distance led to a higher reciprocity ratio with members of the internal network than with the external network. Hence, women were more inclined to utilize the old-age support resources provided by their close members to resolve the old-age support risks. With the increase in age, middle-aged elderly people would reduce their interaction with their secondary partner due to their worldly experience, and to a greater extent trust and rely on the old-age support resources provided by the close internal network members, to reduce the perception of old-age support risks gradually. Our finding supported other studies that education can lead to denser and wider social networks that provide a more comprehensive range of social support, and low levels of education are often accompanied by strained social networks [57,58]. Thus, uneducated older people were more inclined to interact with members of their internal network. The restriction could partly explain the reduction in their perceived level of old-age support risks. As for the communication frequency of external network relationships, the rural elders can use the external network as a bridge to expand external resources. However, the rural elders were still easy affected by the distance in their social stations and preference, causing resistance to external resources. Compared with their poor conditions, superior external resources can be instrumental in reducing the identification of the rural elders, thus improving the perceived level of their old-age support risks. The rural elders mainly rely on the internal network to reduce their perception of old-age support risks. However, whether the internal network effectively reduces their perception of old-age support risks for the rural elders needed to be verified by the latest data. Therefore, we should vigorously explore the important resources contained in the internal network and expand it based on cultivating the rural elders’ internal network. At the same time, attention should be paid to the potential of external networks and the improvement in old-age support resources through the bridge of external networks.

Finally, as a part of the external network, network heterogeneity (most of them are members of the external network) had a significant positive impact on the perceived level of old-age support risks of the rural elders. It could be argued that, as the rural elders had upward comparison behavior with network members when interacting with them, the gap between their high old-age support expectation and reality made a low sense of identity with network members. They were more likely to be reluctant to use their social network resources, leading to a higher level of perception of old-age support risks. In the results of clustering estimation, network heterogeneity had a significant positive effect mainly on middle-aged and educated rural elders. We noted that there may be some age differences in the network; as mentioned above, middle-aged rural elders would increase their trust in close network members and reduce their identification with secondary partners due to worldly experience with the increase in age. The increase in network heterogeneity would also increase their perception of old-age support risks. McPherson et al. proposed that a high-level educational attainment was a positive driver of having larger social networks resulting in a higher degree of network heterogeneity [58]. In addition, educational attainment had stimulated upward comparison behavior among rural elders. Our study suggested a program of educational and cultural interventions to improve internet literacy and reduce the perceived risk of old-age support among older adults. Although rural elders mainly relied on internal networks to reduce their perception of old-age support risks, against the background of rapid urbanization development and most rural adult children going out for work, relying on external networks to develop mutual care for elders can be an effective method to reduce their perception of old-age support risks. Therefore, our study suggested that more support should be given to expand the interpersonal network, promote the concept of mutual support, and encourage the rural elderly to use external network resources to help each other.

This study also had some limitations, mainly reflected in the following aspects. First, limited by human and material resources, the final effective sample size of this study was limited to 1126, which may have led to some deviation in the regression results, and some biases were still inevitable in thoroughly verifying the hypotheses proposed in this study. Secondly, the social network selected in this study was the individual central network of the rural elders, without collecting relevant information on the overall network. Future studies including the overall network would be concluded to reveal the structure of the social network fully. Thirdly, the cross-sectional design of this study could not infer the causal relationships. Future studies are required to identify the causal inference for specific key circumstances of the social network mechanism.

## 5. Conclusions

The OLS regression model, cluster estimation, and robustness test offered substantial evidence to advance our understanding of the social network mechanisms and develop effective strategies for reducing risk. Our study found that a larger internal network size has a significantly protective effect in reducing the old-age support risks of rural elders, while the external network size had no significant impact, so as to answer the research questions and to achieve the aim of our study. In addition, the network heterogeneity and the communication frequency of external network relationships were found to be significantly positively associated with the perception of old-age support risks of rural elders, while the ratio of communication frequency between internal and external network relationships was found to be significantly negatively associated with the perception of old-age support risks of rural elders. This study argued that decision-makers should focus on social network mechanisms, especially the internal network size, the network heterogeneity, the communication frequency of external network relationships, and the ratio of communication frequency between internal and external network relationships, and develop effective targeted measures to reduce the perception of old-age support risks of rural elders.

## Figures and Tables

**Table 1 ijerph-19-15289-t001:** Measurement of network heterogeneity variables.

Number	Problem
1	If you and your spouse or children have a serious conflict or even a quarrel, who usually helps to resolve the conflict?
2	If you were depressed and wanted to talk to someone, who would you talk to about these issues?
3	If you need advice on a major issue in your life, who did you turn to for advice?
4	If you need help with something in your house, who will you ask? (carrying furniture, big bags of grain, heavy objects, etc.)
5	If you or someone in your family is seriously ill and bedridden or needs to be sent to the hospital, who will you ask to take care of you or help you with the housework?
6	If you needed to borrow a tool, who would you ask?
7	If you needed to borrow some money, who would you ask?
8	Who would you ask to help if you had a problem filling out this form?
9	If you need to go out with someone these days, who would you choose?
10	Who do you socialize with at least once a month, such as drinking, visiting, chatting and playing cards?
11	If you need to cooperate with others economically, who would you most like to partner with?
12	If a member of your family goes out to work or goes to the hospital, who has ever helped you?

**Table 2 ijerph-19-15289-t002:** Descriptive statistical analysis results of variables.

Variables	Number/Mean	Percentage/SD
Dependent variable		
Perception of old-age support risks index	57.541	31.100
Independent variables		
Network size	16.274	9.946
Internal network size	9.457	5.250
External network size	6.817	6.514
Network heterogeneity	4.490	2.022
The communication frequency of internal network relationships	10.365	1.561
The communication frequency of external network relationships	9.835	1.935
The ratio of communication frequency between internal and external network relationships	1.091	0.263
Control Variables		
Gender		
Male	521	46.3%
Female †	605	53.7%
Age	69.716	6.664
Self-rated health status		
Very poor †	53	4.7%
Poor	216	19.2%
Fair	379	33.7%
Good	384	34.1%
Very good	94	8.3%
Mental health status		
Very poor †	29	2.6%
Poor	164	14.6%
Fair	375	33.3%
Good	273	24.2%
Very good	285	25.3%
Changes in health status		
Worse †	468	41.6%
Almost unchanged	586	52.0%
Better	72	6.4%
Years of education	3.895	3.542
Mobile phone use		
No †	247	21.9%
Yes	879	78.1%
Pension benefits evaluation		
Very poor †	320	28.4%
Poor	355	31.5%
Fair	353	31.3%
Good	84	7.5%
Very good	14	1.2%
Annual household income level (logarithm of income)	9.131	0.944
Number of children	3.036	1.212
Way of living		
Live alone †	177	15.7%
Live alone with spouse	515	45.7%
Living with children, etc.	434	38.5%
Whether the village has old-age support service places		
None/Unclear †	945	83.9%
Have	181	16.1%
The terrain of the village		
Hills/Mountains †	512	45.5%
Plain	614	54.5%
Region		
West †	621	55.2%
East	232	20.6%
Middle	273	24.2%

Note: † Reference levels in the regressions; SD, standard deviation.

**Table 3 ijerph-19-15289-t003:** Results of the impact of social networks on the perception of old-age support risks of rural elders.

Variables	Model 1	Model 2	Model 3	Model 4
Network size	−0.548 *** (0.104)		−0.421 *** (0.098)	
Internal network size		−1.116 *** (0.198)		−1.029 *** (0.195)
External network size		−0.117 (0.165)		0.033 (0.159)
Network heterogeneity	1.370 *** (0.453)	1.320 *** (0.451)	1.478 *** (0.453)	1.435 *** (0.450)
The communication frequency of internal network relationships	−0.209 (0.672)	0.171 (0.678)		
The communication frequency of external network relationships	2.545 *** (0.558)	2.088 *** (0.572)		
The ratio of communication frequency between internal and external network relationships			−11.440 *** (3.459)	−8.884 ** (3.513)
Control Variables				
Gender				
Female †				
Male	−4.330 ** (1.892)	−4.246 ** (1.883)	−4.159 ** (1.901)	−4.078 ** (1.891)
Age	−0.292 * (0.160)	−0.311 * (0.159)	−0.334 ** (0.160)	−0.357 ** (0.159)
Self-rated health status				
Very poor †				
Poor	−4.118 (4.637)	−4.171 (4.615)	−4.749 (4.655)	−4.830 (4.630)
Fair	−5.530 (4.507)	−5.690 (4.486)	−5.620 (4.530)	−5.788 (4.506)
Good	−14.961 *** (4.604)	−14.579 *** (4.584)	−15.243 *** (4.628)	−14.825 *** (4.604)
Very good	−10.186 * (5.436)	−9.633 * (5.413)	−9.738 * (5.464)	−9.190 * (5.437)
Mental health status				
Very Poor †				
Poor	−8.041 (6.072)	−8.280 (6.044)	−7.396 (6.102)	−7.696 (6.069)
Fair	−10.245 * (5.865)	−9.925 * (5.838)	−9.892 * (5.892)	−9.488 (5.861)
Good	−9.701 (5.980)	−10.281 * (5.955)	−9.014 (6.006)	−9.596 (5.976)
Very good	−13.702 ** (5.980)	−13.773 ** (5.952)	−13.296 ** (6.010)	−13.338 ** (5.977)
Changes in health status				
Worse †				
Almost unchanged	2.141 (1.988)	2.192 (1.979)	1.966 (1.988)	2.022 (1.987)
Better	2.857 (3.735)	2.896 (3.717)	2.675 (3.756)	2.750 (3.736)
Years of education	0.185 (0.276)	0.168 (0.275)	0.160 (0.277)	0.147 (0.276)
Mobile phone use				
No †				
Yes	−3.153 (2.286)	−3.408 (2.277)	−3.159 (2.297)	−3.467 (2.286)
Pension benefits evaluation				
Very poor †				
Poor	−6.101 *** (2.275)	−6.123 *** (2.265)	−5.771 ** (2.284)	−5.851 ** (2.272)
Fair	−13.592 *** (2.300)	−13.433 *** (2.290)	−13.958 *** (2.310)	−13.794 *** (2.298)
Good	−12.638 *** (3.729)	−12.569 *** (3.711)	−12.194 *** (3.748)	−12.174 *** (3.728)
Very good	−28.110 *** (8.224)	−28.885 *** (8.189)	−28.740 *** (8.261)	−29.643 *** (8.220)
Annual household income level (Logarithm of income)	−1.517 (1.047)	−1.617 (1.042)	−1.216 (1.049)	−1.335 (1.043)
Number of children	1.615 * (0.840)	1.913 ** (0.841)	1.743 ** (0.843)	2.070 ** (0.843)
Way of living				
Living alone †				
Live alone with spouse	−0.952 (2.665)	−0.437 (2.657)	−0.700 (2.676)	−0.113 (2.667)
Living with children, etc.	−4.422 (2.703)	−2.965 (2.725)	−4.616 * (2.714)	−2.990 (2.737)
Whether the village has old-age support service places				
None/Unclear †				
Have	−5.653 ** (2.457)	−5.902 ** (2.446)	−5.307 ** (2.469)	−5.615 ** (2.457)
The terrain of the village				
Hills/Mountains †				
Plain	−4.045 ** (1.914)	−3.861 ** (1.906)	−3.719 * (1.922)	−3.573 * (1.912)
Region				
West †				
East	−5.434 ** (2.475)	−4.762 * (2.471)	−6.070 ** (2.477)	−5.363 ** (2.472)
Middle	−3.521 (2.401)	−3.735 (2.391)	−3.551 (2.414)	−3.782 (2.401)
Constant	1020.886 *** (16.512)	106.619 *** (16.472)	135.216 *** (15.828)	136.095 *** (15.744)
R^2^	0.180	0.188	0.170	0.180

† Reference levels in the regressions; Standard error in brackets, * *p* < 0.1, ** *p* < 0.05, *** *p* < 0.01.

**Table 4 ijerph-19-15289-t004:** Regression results of social networks’ clustering estimation of rural elders’ perception of old-age support risks.

Variables	Model 5 (Male)	Model 6 (Female)	Model 7 (Young Elders) (Age 60–69)	Model 8 (Middle-Aged Elders) (Age 70+)	Model 9 (Uneducated)	Model 10 (Be Educated)
Internal network size	−1.466 *** (0.291)	−0.673 ** (0.270)	−1.318 *** (0.270)	−0.569 ** (0.283)	−0.853 ** (0.339)	−1.164 *** (0.241)
External network size	0.095 (0.244)	−0.022 (0.212)	−0.065 (0.221)	0.233 (0.232)	−0.172 (0.290)	0.155 (0.193)
Network heterogeneity	1.943 *** (0.650)	1.128 * (0.640)	0.914 (0.627)	1.765 *** (0.657)	1.103 (0.868)	1.738 *** (0.539)
The ratio of communication frequency between internal and external network relationships	−3.857 (6.046)	−12.114 *** (4.393)	−7.840 (5.698)	−8.528 * (4.476)	−14.592 *** (5.262)	−4.124 (4.821)
Control Variables	Have control	Have control	Have control	Have control	Have control	Have control
Constant	99.172 *** (25.671)	153.742 *** (20.600)	84.882 *** (16.928)	140.478 *** (17.468)	120.521 *** (24.476)	133.342 *** (21.512)
R^2^	0.226	0.185	0.187	0.240	0.226	0.199

Note: Standard error in brackets, * *p* < 0.1, ** *p* < 0.05, *** *p* < 0.01.

**Table 5 ijerph-19-15289-t005:** Robustness test results (Substitution of explained variables).

Variables	Model 11	Model 12	Model 13	Model 14
Network size	−0.066 *** (0.013)		−0.051 *** (0.012)	
Internal network size		−0.136 *** (0.024)		−0.125 *** (0.024)
External network size		−0.014 (0.020)		0.004 (0.019)
Network heterogeneity	0.166 *** (0.055)	0.160 *** (0.055)	0.179 *** (0.055)	0.174 *** (0.055)
The communication frequency of internal network relationships	−0.025 (0.082)	0.022 (0.082)		
The communication frequency of external network relationships	0.308 *** (0.068)	0.252 *** (0.069)		
The ratio of communication frequency between internal and external network relationships			−1.385 *** (0.420)	−1.074 ** (0.426)
Control Variables	Have control	Have control	Have control	Have control
Constant	15.295 *** (2.003)	15.751 *** (1.998)	19.213 *** (1.920)	19.320 *** (1.910)
R^2^	0.179	0.187	0.170	0.179

Note: Standard error in brackets, ** *p* < 0.05, *** *p* < 0.01.

## Data Availability

The datasets generated and analyzed during the current study are available from the corresponding author upon reasonable request.

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
