# Peer review of "By Internal Network or by External Network?—Study on the Social Network Mechanism of Reducing the Perception of Old-Age Support Risks of Rural Elders in China"

_ijerph, 2022, doi:10.3390/ijerph192215289_

Round 1

Reviewer 1 Report

Overall, the subject of the study is interesting and relevant from a practical and research point of view. The summary summarizes the main ideas of the study. The introduction is well laid out, establishing an adequate starting point for the study. The objectives are well stated. Regarding the methodology, the sample selection and the different steps of the research should be better explained. Differences regarding gender are presented, but the correct approach would be to establish differences regarding sex (male vs. female), since gender is a social and cultural construct. The sample size is ample, which is a strength for the study. Likewise, the statistical analyses used should be better justified. The results are well presented. In the discussion section, the results obtained are correctly argued. In the conclusions section, the practical relevance of the study and the novelty of the research with respect to previous studies should be discussed in greater depth. As for the references, a considerable number are presented for an academic work, but they should be updated as far as possible.
As for the format, the work presents a good structure, but the writing is sometimes a bit confusing, the formal language is adequate for an academic work. I suggest you to lighten the wording of some sections and maintain coherence between the different paragraphs of the text.
Finally, I encourage them to continue in this line of research.
My best wishes to the authors.

Author Response

We thank the reviewer for the careful reading and constructive comments on our manuscript. We have made every effort to revise our paper in light of the comments. The associated changes are highlighted in red color in the revised manuscript, and the following is our detailed responses to the comments (in order of occurrence).

Point 1: Overall, the subject of the study is interesting and relevant from a practical and research point of view. The summary summarizes the main ideas of the study. The introduction is well laid out, establishing an adequate starting point for the study. The objectives are well stated. The sample size is ample, which is a strength for the study. The results are well presented. In the discussion section, the results obtained are correctly argued.

Response 1:

Thank you very much for your recognition of our study. We really appreciate your interest in our research on the social networks on the perception of old-age support risks of rural elders in China.

Point 2: Regarding the methodology, the sample selection and the different steps of the research should be better explained.

Response 2:

Thank you for the constructive comment. We apologize for not making a detailed introduction about the sample selection and the different steps of the research. We have made some adjustments in the revised manuscript following your advice. Please check and revise(can be found in line 117 to 133, page 3).

The following are the corresponding revisions:

The data was collected from the sample survey of older adults aged 60 and above in 11 representative provinces in eastern, central and western China. The survey was conducted by the Social Security Research Group of Northwest University in China, which is funded by the project of the National Social Science Foundation. The questionnaire was designed based on large-scale scientific surveys and relevant studies at home and abroad, then conducted a trial survey. We modified and optimized the questionnaires according to comprehensive consideration of results of the trial survey, the validity and the reliability. Secondly, the research group randomly selected 32 highly representative villages from 11 provinces according to the socioeconomic development level of different regions, including four provinces in the eastern region (Hebei, Fujian, Jiangsu and Shandong), four provinces in the central region (Shanxi, Henan, Hubei and Hunan), and three provinces in the western region(Shaanxi, Yunnan and Xinjiang). Finally, the survey randomly selected about 35 older adults aged 60 and above from each village, and a questionnaire survey was conducted through face-to-face interview. One of sample villages was a non-traditional village, which can not meet the sampling requirements well, and the actual sample size was small, so 1126 valid questionnaires from 31 villages were retained.

Point 3: Differences regarding gender are presented, but the correct approach would be to establish differences regarding sex (male vs. female), since gender is a social and cultural construct.

Response 3:

Thank you for the constructive comment. It is really true as reviewer suggested differences regarding sex (male vs. female) should be considered. We feel sorry that we didn’t clearly present more information of differences in male, female. We have made some adjustments on the differences regarding male, female, young elders, middle-aged elders, uneducated elders, educated elders in the revised manuscript following your advice. Please check and revise(can be found in line 288 to 310, page 10 to page 11).

The following are the corresponding revisions:

As Table 4 showed, the clustering estimation results of social networks’ perception of old-age support risks for rural elders were reported. Gender (Model 5 and Model 6), age (Model 7 and Model 8) and education (Model 9 and Model 10) were highly related to the individual characteristics of the rural elders, and they were deliberately selected as the group estimation standards. According to international standards and research data, the rural elders between 60 and 69 as the young elders, and farmers aged 70 and above in the village were classified as Middle-age elders [30]. According to the number of years of education, the rural elders were divided into uneducated and educated groups. The size of the internal network could significantly reduce the perception of old-age support risks for males (Model 5) and females (Model 6), young elders (Model 7), middle-age elders (Model 8) and rural elders with various levels of education (Model 9 and Model 10), which indicated that family security was always the primary choice for rural elders. The influence of external network size was not significant among different populations.

However, the network heterogeneity and the ratio of communication frequency between internal and external network relationships had different effects on the perception of old-age support risks of rural elders with different genders (Model 5 and Model 6), ages (Model 7 and Model 8) and educational attainment (Model 9 and Model 10). There was a positive association between network heterogeneity and the variables of males (Model 5), females (Model 6), middle-aged elders (Model 9) and educated rural elders (Model 8). While the ratio of communication frequency between internal and external network relationships had a significant negative influence on females (Model 6), middle-aged elders (Model 8) and uneducated rural elders (Model 9).

Point 4: Likewise, the statistical analyses used should be better justified.

Response 4:

Thank you for the constructive comment. We apologize for not making a detailed introduction about the statistical analyses. We have made some adjustments in the revised manuscript following your advice. Please check and revise(can be found in line 229 to 244, page 7).

The following are the corresponding revisions:

Since the dependent variable, the perception of old-age support risks of elders, was a continuous variable, our study used the ordinary least square (OLS) regression model to analyze the factors affecting the social networks of rural elders. Regression coefficients and standard errors were reported in the regression results. A five percent significance level was chosen as a threshold for the inclusion of the model variables. All statistical analyses were performed using SPSS 26.0. The regression model was set as follows:

In Equation (1), y presented the perception of old-age support risks of elders,  presented the independent variable,  meant the control variable, b1,b2,bn presented regression coefficients respectively,  represented error items. Further, on the basis of the OLS regression model, we discussed the influence of social network and other variables on the perception of old-age support risks in different groups through the clustering estimation. Similarly, our study adopted the method of replacement of the dependent variable for the robustness test in the end, which is convenient for enhancing the explanatory power of the model.

Point 5: In the conclusions section, the practical relevance of the study and the novelty of the research with respect to previous studies should be discussed in greater depth.

Response 5:

Thank you for the constructive comment. We apologize for not making a detailed introduction about the practical relevance of the study and the novelty of the research with respect to previous studies. We clarified the novelty of the research, that is, our study offered substantial evidence to advance our understanding of the social network mechanisms and develop effective strategies for reducing risk. In addition, our study argued that decision-makers should focus on social network mechanisms, and develop effective targeted measures to reduce the perception of old-age support risks of rural elders(can be found on line 461 to line 462, page 15; line 461 to line 462, page 15).

The following are the corresponding revisions:

The OLS regression model, cluster estimation and robustness test offered substantial evidence to advance our understanding of the social network mechanisms and develop effective strategies for reducing risk.

This study argued that decision-makers should focus on social network mechanisms, especially internal network size, the network heterogeneity, the communication frequency of external network relationships, the ratio of communication frequency between internal and external network relationships, and develop effective targeted measures to reduce the perception of old-age support risks of rural elders.

Point 6: As for the references, a considerable number are presented for an academic work, but they should be updated as far as possible.

Response 6:

Thank you for the insightful comment. It is really true as reviewer suggested that our references were a little old, but they should be updated as far as possible. We have renewed some references and added some references in the revised manuscript following your advice. Please check.

The following are the corresponding revisions:

19.Zhang, D. Impacts of Intergenerational Support and Social Network on the Life Quality of the Elderly. Journal of China Institute of Labor Relations 2021, 35, 15-25.

20.Zheng, Z.; Chen, H. Age sequences of the elderly’social network and its efficacies on well-being: an urban-rural comparison in China. BMC Geriatrics 2020, 20, 1-10, https://doi.org/10.1186/s12877-020-01773-8.

26.Wu, Y.L.; Xiao, P.F.; Liu, H.; Dong, Y.B.; Qin, S. Safety Risk Assessment of Street Tree Based on Factor Analysis and k-means Clustering. Journal of Southwest Forestry University 2022, 1-9.

27.Zamprogno, B.; Reisen, V. A.; Bondon, P.; Aranda Cotta, H.H.; Reis Jr, N.C. Principal component analysis with autocorrelated data. Journal of Statistical Computation and Simulation 2020, 90, 2117-2135, https://doi.org/10.1080/00949655.2020.1764556.

28.Zhang,Y.P.; Jin, W.l.; Dong, C.Y.; Niu, J.M.; Wu, Y. Asynchronous Scaled Principal Component Analysis for High-dimensional Time Series Data Prediction. Journal of Shanxi University (Natural Science Edition) 2022, 1-5.

29.Lubben, J.; Blozik, E.; Gillmann, G.; Iliffe, S. von Renteln Kruse; Beck, J,C.; Stuck, A.E. Performance of an Abbreviated Version of the Lubben Social network size among Three European Community-Dwelling Older Adult Populations. The Gerontologist 2006, 503-513.

40.Li, D.; Zhang, J.; Yang, J.; Xu, Y.; Lyu, R.; Zhong, L.; Wang, X.. Socio-economic inequalities in health service utilization among Chinese rural migrant workers with New Cooperative Medical Scheme: a multilevel regression approach[J]. BMC Public Health. 2022 Jun 3;22(1):1110. doi: 10.1186/s12889-022-13486-1.

41.Li, D.; Zhai, S.; Zhang, J.; Yang, J.; Wang X.. Assessing Income-Related Inequality on Health Service Utilization among Chinese Rural Migrant Workers with New Co-operative Medical Scheme: A multilevel Approach[J]. Int. J. Environ. Res. Public Health 2021, 18. doi:10.3390/ijerph182010851.

Response 7:

Thank you for the insightful comment. We apologize for the expression in English in our paper. Given the reviewer have been proposed to revise the papers in English error, we have modified the manuscript in grammar, spelling, sentence structure errors, and strive to achieve the requirements of the journal. Please check.

Special thanks to you for your good comments!

Reviewer 2 Report

The authors are analysing the general trend for China of the deeply aging society. They analysed 1126 rural elders in 11 provinces of China applying descriptive statistics and OLS methods. 

Even if the paper is well-written, it does not present the standard structuration of a high-impact scientific article, because, the introduction has to provide a brief overview of what the paper will treat. Then, it has to follow a literature review of the topic.

Data section. Where did you extract the dataset? You mention that "data was collected from the sample survey of older adults in 31 villages" but it is a survey that you have created or it is a survey of a large database (such as ISSP)?

The paper also mentions the use of principal component analysis and factor analysis but does not introduce these methods. A no-expert on these approaches does not understand what are you doing. 

Results, Discussions are well-written and clearly presented. The conclusion should contain the main goal reached with this study. Does the methodology provide efficient results? And, mainly, it has to contain the answer to the main research question. 

Minor comments:

- Abstract has not to be subdivided into sections. 

Author Response

We thank the reviewer for the careful reading and constructive comments on our manuscript. We have made every effort to revise our paper in light of the comments. The associated changes are highlighted in red color in the revised manuscript, and the following is our detailed responses to the comments (in order of occurrence).

Point 1: The authors are analysing the general trend for China of the deeply aging society. They analysed 1126 rural elders in 11 provinces of China applying descriptive statistics and OLS methods. Results, Discussions are well-written and clearly presented.

Response 1:

Thank you very much for your recognition of our study. We really appreciate your interest in our research on the social networks on the perception of old-age support risks of rural elders in China.

Point 2:The introduction has to provide a brief overview of what the paper will treat. Then, it has to follow a literature review of the topic.

Response 2:

Thank you for the constructive comment. We apologize for not making a brief overview of what the paper will treat. We have made some adjustments(can be found in line 109 to 112, page 3). and added a literature review of the topic in the revised manuscript following your advice. Please check.

The following are the corresponding revisions:

Therefore, this paper used the questionnaire data of 1126 rural elders in 11 provinces of China to explore the reduction effect of social networks on the perception of old-age support risks of rural elders from the perspective of the size, heterogeneity and communication frequency of social network(can be found in line 117 to 133, page 3).

In addition, many researches have proved that intimate interactions between older adults and members of a social network contributed to relieving stress and promoting physical and mental health [17,18], and in turn improved their quality of life [19]. Several studies concluded that family network in social networks had the "reverse structure " in age sequences and has very important effects the well-being of the rural elderly [20]. Many scholars have pointed out that rural elders with diversified social networks had lower levels of depressive symptoms [21,22]. In terms of social network mechanism, evidences have demonstrated the interrelationships of both the causal mechanisms by which social network influences physical health and the ways in which this influence can inform potential intervention strategies [23]. However, few scholars have studied the association between social network mechanism and the perception of old-age support risk of rural elders(can be found in line 93 to 104, page 2 to page 3).

Point 3: Data section. Where did you extract the dataset? You mention that "data was collected from the sample survey of older adults in 31 villages" but it is a survey that you have created or it is a survey of a large database (such as ISSP)?

Response 3:

Thank you for the constructive comment. We apologize for not making a detailed introduction about the dataset we used. We have made some adjustments in the revised manuscript following your advice. Please check (can be found in line 117 to 133, page 3).

The following are the corresponding revisions:

The data was collected from the sample survey of older adults aged 60 and above in 11 representative provinces in eastern, central and western China. The survey was conducted by the Social Security Research Group of Northwest University in China, which is funded by the project of the National Social Science Foundation. The questionnaire was designed based on large-scale scientific surveys and relevant studies at home and abroad, then conducted a trial survey. We modified and optimized the questionnaires according to comprehensive consideration of results of the trial survey, the validity and the reliability. Secondly, the research group randomly selected 32 highly representative villages from 11 provinces according to the socioeconomic development level of different regions, including four provinces in the eastern region (Hebei, Fujian, Jiangsu and Shandong), four provinces in the central region (Shanxi, Henan, Hubei and Hunan), and three provinces in the western region(Shaanxi, Yunnan and Xinjiang). Finally, the survey randomly selected about 35 older adults aged 60 and above from each village, and a questionnaire survey was conducted through face-to-face interview. One of sample villages was a non-traditional village, which can not meet the sampling requirements well, and the actual sample size was small, so 1126 valid questionnaires from 31 villages were retained.

Point 4: The paper also mentions the use of principal component analysis and factor analysis but does not introduce these methods. A no-expert on these approaches does not understand what are you doing.

Response 4:

Thank you for the constructive comment. We apologize for not making a detailed introduction about principal component analysis and factor analysis. We have made some adjustments in the revised manuscript following your advice. Please check(can be found in line 149 to 157, page 3 to page 4).

The following are the corresponding revisions:

In order to measure the perceived degree of old-age support risks of the rural elders on the whole, the Principal Component Analysis (PCA) was used to carry out factor analysis on these three dimensions and extract a common factor. Factor analysis is a kind of multivariate statistical analysis method in which some variables with complex relationships are attributed to a few unrelated comprehensive factors based on the internal dependence of the index correlation matrix [26]. PCA is a statistical method for data dimensionality reduction [27]. It converts high-dimensional variables into low-dimensional orthogonal variables, and the converted variables are called principal components [28]. PCA can provide initial solutions for factor analysis.

Point 5: The conclusion should contain the main goal reached with this study. Does the methodology provide efficient results? And, mainly, it has to contain the answer to the main research question.

Response 5:

Thank you for the constructive comment. We apologize for not making a detailed introduction about the main goal and main research question in the conclusion. We have renewed the conclusion in the revised manuscript following your advice. Please check (can be found in line 461 to 477, page 15).

The following are the corresponding revisions:

The OLS regression model, cluster estimation and robustness test offered substantial evidence to advance our understanding of the social network mechanisms and develop effective strategies for reducing risk. Our study found that larger internal network size has a significantly protective effect in reducing old-age support risks of rural elders, while the external network size has no significant impact, so as to answer the research questions and to achieve the aim of our study. In addition, the network heterogeneity and the communication frequency of external network relationships were found to be significantly positively associated with the perception of old-age support risks of rural elders, while the ratio of communication frequency between internal and external network relationships was found to be significantly negatively associated with the perception of old-age support risks of rural elders. This study argued that decision-makers should focus on social network mechanisms, especially internal network size, the network heterogeneity, the communication frequency of external network relationships, the ratio of communication frequency between internal and external network relationships, and develop effective targeted measures to reduce the perception of old-age support risks of rural elders.

Special thanks to you for your good comments!

Round 2

Reviewer 2 Report

Thank you for having taken my comments into consideration. The new version of the paper is undoubtedly better. Nonetheless, the literature review is still poor. It is very a very important section for scientific papers! Maybe authors can enrich this part and create a proper theoretical background section. 

Author Response

Thank you for your letter and for the reviewers’ comments concerning our manuscript entitled “By Internal Network or by External Network?——Study on the Social Network Mechanism of Reducing the Perception of Old-age Support Risks of Rural Elders in China” (ID: ijerph-2018607). We have studied comments carefully and have made correction which we hope meet with approval. Revised portion are marked in red in the paper.

Reviewer #2: Nonetheless, the literature review is still poor. It is very a very important section for scientific papers! Maybe authors can enrich this part and create a proper theoretical background section.

Response: Special thanks to you for your good comment. The comments are valuable and very helpful for revising and improving our paper. We have made some adjustments in the revised manuscript following your advice. Please check (can be found in line 61 to 64, page 2; line 79 to 82, page 2; line 98 to 115, page 3; line 120 to 126, page 3).

The following are the corresponding revisions:

Plenty of problems may be more common among the rural elders who face physical disability, psychological stress, and economic cost barriers caused by spatial transformation, accompanied by the characteristics of increased health risks as they age, low education levels, low income, and lack of companionship.

However, there is a relative lack of research regarding the social network mechanism of rural elders in China. More attention is required, in particular, the relationship between social network mechanism and their perception of old-age support risks.

Social network can be referred to as a composite characterization of the main components of an individual's social network relationships and a comprehensive description of their interpersonal environment [17]. The construct of social network can be generally defined as social capital, reflects the collection of interpersonal contacts that people variously maintain and which provide them with access to social, emotional and practical support [18]. Studies have reported on the structural attributes (e.g., network member's composition, size and contact frequency) or measurement scales (e.g., ‘family’, ‘friends’, ‘diverse’ and ‘restricted’ networks) to identify different social network [19, 20]. Social networks could be divided into family networks and friend networks based on kinship [21]. Among them, family network had the "reverse structure " in age sequences and has very important effects the well-being of the rural elderly [22]. Family networks could also improve life satisfaction for elderly by decreasing the level of physical abuse they experience [23]. Older adults' interactions with their friend networks could enhance their sense of self-worth, improve negative psychological [24], and enhance the health status [25]. In addition, many researches have proved that intimate interactions between older adults and members of a social network contributed to relieving stress, reducing depression levels and promoting physical and mental health [26-28], and in turn improved their quality of life [30,31]. 

In the field of old-age support risk, most studies mainly focused on the overall old-age support risks faced by the elderly [6,7,36]. Only a few studies have explored its impact on perception of old-age support risk from the perspectives of family security [37], social security [38] and offspring support [39]. However, evidence on the association between social network mechanism and the perception of old-age support risk among rural elders still lacked support from a wider range of data.

References

  1. Chen, J.S. New risks and management of rural endowment in China. Journal of Hainan University (Humanities and Social Sciences) 2020,38, 59-65, https://doi.org/10.15886/j.cnki.hnus.2020.05.009.
  2. Mu, G.J. The Risk of Elderly Support in the Era of Low-fertility. Journal of Huazhong University of Science and Technology (Social Science Edition) 2018, 32, 1-7, https://doi.org/10.19648/j.cnki.jhustss1980.2018.01.003.
  3. Litwin H, Stoeckel KJ. Social networks and subjective wellbeing among older Europeans: does age make a difference? Ageing Soc. 2013; 33(7): 1263-1281.https://doi.org/10.1017/S0144686X12000645
  4. Gray A . The social capital of older people[J]. Ageing & Society, 2009, 29(1):5-31. https://doi.org/10.1017/S0144686X08007617
  5. Park NS, Jang Y, Lee BS, et al. Associations of a social network typology with physical and mental health risks amongolder adults in South Korea. Aging Ment Health. 2018;22(5): 631-638. https://doi.org/10.1080/13607863.2017.1286456
  6. 20. Li T, Yang YC, Zhang Y. Culture, economic development, social‐network type, and mortality: evidence from Chineseolder adults. SocSci  2018; 204: 23-30. https://doi.org/10.1016/j.socscimed.2018.03.021
  7. Lubben, J.; Blozik, E.; Gillmann, G.; Iliffe, S.;von Renteln Kruse, W.; Beck, J.C.; Stuck, A.E. Performance of an Abbreviated Version of the Lubben Social network size among Three European Community-Dwelling Older Adult Populations. The Gerontologist 2006, 46, 503-513, https://doi.org/10.1093/geront/46.4.503.
  8. 22.Zheng,Z.; Chen,  Age sequences of the elderly’social network and its efficacies on well-being: an urban-rural comparison in China. BMC Geriatrics 2020, 20, 1-10, https://doi.org/10.1186/s12877-020-01773-8.
  9. Cheng, W.; Song, W.; Ye, C.; Wang, Z. Family Networks, Social Networks, and Life Satisfaction of Older Adults in China. Health Care 2022, 10, 1568, https://doi.org/10.3390/healthcare10081568.
  10. Tang, D.; Lin, Z.; Chen, F. Moving beyond living arrangements: the role of family and friendship ties in promoting mental health for urban and rural older adults in China. Aging & Mental Health 2020, 24, 1523-1532, https://doi.org/10.1080/13607863.2019.1602589.
  11. Park, N.S.; Jang, Y.; Lee, B.S.; Chiriboga, D.A.; Chang, S.; Kim, S.Y. Associations of a social network typology with physical and mental health risks among older adults in South Korea. Aging & Mental Health 2017, 22, 631-638, https://doi.org/10.1080/13607863.2017.1286456.
  12. 26.Santini, Z.I.; Koyanagi, A.; Tyrovolas, S.; Mason, C.; Haro, J.M. The association between social relationships and depression: a systematic review. Journal of Affective Disorders 2015, 53-65, https://doi.org/10.1016/j.jad.2014.12.049.
  13. 27.Litwin, H.; Stoeckel, K.J.; Schwartz, E. Social networks and mental health among older Europeans: are there age effects?. European Journal of Ageing 2015, 12, 299-309,https://doi.org/10.1007/s10433-015-0347-y.
  14. Domènech-Abella, J.; Lara, E.; Rubio-Valera, M.; Olaya, B.; Moneta, M.V.; Rico-Uribe, L.A.; Ayuso-Mateos, J.L.; Mundó, J.; Haro, J.M. Loneliness and depression in the elderly: the role of social network. Social Psychiatry and Psychiatric Epidemiology 2017, 52, 381-390, https://doi.org/10.1007/s00127-017-1339-3.
  15. Lei, P.; Xu, L.; Nwaru, B.I.; Long, Q.; Wu, Z. Social networks and health-related quality of life among Chinese old adults in urban areas: results from 4th National Household Health Survey. Public health 2016, 131, 27-39, https://doi.org/10.1016/j.puhe.2015.10.009.
  16. 31.Zhang, D. Impacts of Intergenerational Support and Social Network on the Life Quality of the Elderly.Journal of China Institute of Labor Relations 2021, 35, 15-25.
  17. Cohn-Schwartz, E.; Levinsky, M.; Litwin, H. Social network type and subsequent cognitive health among older Europeans. International Psychogeriatrics 2021, 33, 495-504, https://doi.org/10.1017/S1041610220003439.
  18. Litwin, H.; Levinsky, M.; Schwartz, E. Network type, transition patterns and well-being among older Europeans. Eur J Ageing 2020, 17, 241-250, https://doi.org/10.1007/s10433-019-00545-7.
  19. Choi, K.-W.; Jeon, G.-S. Social Network Types and Depressive Symptoms among Older Korean Men and Women. Int. J. Environ. Res. Public Health 2021, 18, 11175, https://doi.org/10.3390/ijerph182111175.
  20. 35.Holt-Lunstad, J. Why Social Relationships Are Important for Physical Health: A Systems Approach to Understanding and Modifying Risk and Protection. Annual Review of Psychology 2018, 69, 437-458, https://doi.org/10.1146/annurev-psych-122216-011902.
  21. Chen, X.; Chen D.Y. Shift of Focus in China’s Population Policy: From Population Control to Old-age Risk Mitigation. Journal of Jinan University (Social Science Edition) 2021, 31, 28-42+173.
  22. Nie, J.L.; Fan, R. By Spouse or Children? -- Study on Family Security Mechanism of Rural Elderly’s Risk Perception Reduction. Journal of Huazhong University of Science and Technology (Social Science Edition) 2021, 35, 56-65, https://doi.org/10.19648/j.cnki.jhustss1980.2021.06.07.
  23. Nie, J.L.; Zhong, Z.B. Family Security, Social Security and Farmers' Worry about Pension -- Based on the empirical investigation of Kong Town, Hubei Province. Rural economy 2014, 90-94.
  24. Hao, J.; Wang, W. The impact of Child Support on pension worries: Based on the 2015 first-generation One-child Family Survey. The World of Survey and Research 2017, 13-16+21, https://doi.org/10.13778/j.cnki.11-3705/c.2017.07.003.

Special thanks to you for your good comments!
